# Nationwide Screening Practices for Tamoxifen Retinal Toxicity in South Korea: A Population-Based Cohort Study

**DOI:** 10.3390/jcm13082167

**Published:** 2024-04-09

**Authors:** Seong Joon Ahn, Jiyeong Kim, Hyeon Yoon Kwon

**Affiliations:** 1Department of Ophthalmology, Hanyang University Hospital, Hanyang University College of Medicine, 222-1 Wangsipli-ro, Seongdong-gu, Seoul 04763, Republic of Korea; 2Department of Pre-Medicine, College of Medicine, and Biostatistics Lab, Medical Research Collaborating Center (MRCC), Hanyang University, Seoul 04763, Republic of Korea

**Keywords:** drug usage, retinal toxicity, screening practices, tamoxifen

## Abstract

**(1) Background/Objectives:** To investigate the nationwide screening practices and trends in tamoxifen retinal toxicity (tamoxifen retinopathy) in South Korea using national health insurance claims data. **(2) Methods:** A total of 43,848 patients who started tamoxifen therapy between 2015 and 2020 and had no prior ophthalmic diseases or other conditions requiring screening for retinopathy were included. The annual numbers of tamoxifen users and new initiators of tamoxifen therapy were assessed. The screening examinations were separated into baseline (first ophthalmic examination after tamoxifen administration) and subsequent monitoring examinations. The timing and modalities for the baseline and subsequent monitoring examinations performed between 2015 and 2021 were assessed in tamoxifen users. **(3) Results:** The annual number of tamoxifen users increased over the study period from 54,056 in 2015 to 81,720 in 2021. The number of patients who underwent ophthalmic examination after tamoxifen administration was 8961 (20.4%). Baseline screening was performed in 6.5% of patients within 1 year of use, and subsequent monitoring was performed in 27.8% of patients who underwent baseline screening. Funduscopy or fundus photography was performed most commonly for baseline screening and subsequent monitoring (99.0% and 98.6%, respectively), while optical coherence tomography was performed only in 21.9% and 29.6% of baseline and monitoring examinations, respectively. The average number of monitoring examinations per year was 0.68 ± 0.45. Although the annual percentage of patients receiving a baseline examination within 1 year gradually increased over time, the percentage of those with subsequent monitoring performed within 1 year was similar over the study period. **(4) Conclusions:** Our finding, appropriate screening in a small proportion of patients receiving tamoxifen, suggests the need to promote awareness among healthcare professionals and develop a standardized approach for screening for tamoxifen retinopathy.

## 1. Introduction

Tamoxifen, a selective estrogen receptor (ER) modulator, is widely used to treat and prevent hormone-receptor-positive breast cancer [1,2]. It has proven effective in reducing the risk of cancer recurrence and improving survival rates in women with ER-positive breast tumors [3,4]. However, long-term use of tamoxifen has been associated with ocular side effects, including tamoxifen retinopathy, a condition that affects the retina and potentially leads to vision loss [1,5,6].

Tamoxifen retinal toxicity, namely, tamoxifen retinopathy, typically manifests as a bilateral symmetrical condition, although asymmetrical involvement can also occur. The clinical features of tamoxifen retinopathy vary from mild to severe. The common signs of tamoxifen retinopathy include crystalline deposits within the macula, known as refractile bodies or tamoxifen crystals [1]. These deposits are often observed in the inner retinal layers and can be visualized by fundus examination. As the condition progresses, other characteristic findings may emerge, such as retinal pigment epithelial changes including granularity, mottling, and pigment clumping. Pseudocystic macular changes and edema may develop, leading to visual disturbances [2]. In advanced stages of tamoxifen retinopathy, retinal atrophy and optic disc pallor may be observed [7].

Accordingly, the timely detection of tamoxifen retinopathy is crucial for implementing appropriate interventions to prevent irreversible structural and functional (vision) loss [1]. Unfortunately, recommendations have not yet been provided by expert panels or organizations, nor has there been a consensus on the screening frequency and modalities for tamoxifen retinopathy. Nationwide practice patterns for tamoxifen retinopathy screening have not yet been reported.

This study intended to investigate practice patterns for screening for tamoxifen retinopathy in Korean population. We aimed to explore the timing and diagnostic modalities employed in screening for retinopathy and highlight the challenges and limitations of real-world practice in Korea.

## 2. Materials and Methods

### 2.1. Study Population

The participants in this cohort study were identified using the Health Insurance Review and Assessment database, which holds comprehensive health claims data for approximately 50 million individuals in South Korea. This database includes detailed records of medication prescriptions, visit dates, and demographic information, along with diagnoses categorized according to the Korean Standard Classification of Diseases, 8th Revision, with adaptations based on the International Statistical Classification of Diseases and Related Health Problems, Tenth Revision (ICD-10). From the database, we identified tamoxifen users who were treated with tamoxifen between 1 January 2012 and 31 December 2021. Patients who had used tamoxifen before 1 January 2015 were excluded to obtain the population who started the therapy between 2015 and 2021 (and thus their treatment duration could be accurately assessed within the study period).

Additionally, individuals who had undergone ophthalmic examinations, including fundus photography or optical coherence tomography (OCT), for any preexisting ophthalmic disease (ICD-10 codes H00-H59) before the initiation of tamoxifen treatment or those with diabetes mellitus or common ophthalmic diseases were excluded to eliminate visits scheduled for monitoring preexisting or common eye diseases or diabetic retinopathy. Further details of the inclusion and exclusion criteria, as well as the number of patients, are presented in Figure 1.

The study was approved by the Institutional Review Board of Hanyang University Hospital (IRB file no. 2023-01-003) and conducted in accordance with the principles of the Declaration of Helsinki. The need for informed consent was waived by the Institutional Review Board of Hanyang University Hospital because of the retrospective nature of the study and the use of deidentified data.

### 2.2. Definitions and Evaluations

Several definitions are used in this study. Baseline examination was defined as the first ophthalmic examination performed after the initiation of tamoxifen therapy as the initial assessment of retinal toxicity associated with tamoxifen use. Subsequent monitoring was defined as examinations performed after baseline examination [8]. 

Several parameters and outcome measures were evaluated as follows. First, we evaluated the annual number of tamoxifen users from 2015 to 2021. This provided the trends and changes in total and new tamoxifen users. Second, the timing of the examinations, baseline or subsequent monitoring, was assessed. Together with the interval between the start date of tamoxifen use and baseline examination, and that between baseline and subsequent monitoring examinations, we examined the frequency of patients who underwent baseline screening within 1 year of initiating tamoxifen therapy, as well as the percentage of patients receiving subsequent monitoring within 6 months and 1 year from the time of baseline screening. The modalities used for baseline screening and subsequent monitoring examinations were also documented. We specifically recorded the use of OCT, funduscopy/fundus photography, automated visual field (VF), fundus autofluorescence (FAF), and fluorescein angiography (FA) for both baseline and monitoring examinations to obtain information on the preferred diagnostic techniques for screening for tamoxifen-induced retinal toxicity. Finally, the number of monitoring examinations per year was calculated to assess the frequency at which patients underwent subsequent monitoring examinations.

### 2.3. Data Analysis

This study employed descriptive statistics to summarize and present the findings. Categorical variables are presented as frequencies and percentages, while continuous variables are reported as mean (standard deviation) or median (interquartile range) values. Fisher’s exact or Chi-square tests were used to compare categorical variables between groups. All *p*-values are based on two-sided tests, and statistical significance was considered at *p* < 0.05. Statistical analyses were conducted using SAS Enterprise Guide version 7.1 (SAS Institute, Cary, NC, USA).

## 3. Results

### 3.1. Population of Tamoxifen Users and Trends over Time

In our study, we identified 43,848 tamoxifen users without prior ophthalmic diseases or conditions requiring retinopathy screening, of whom 88.6% were female. The demographic and clinical characteristics of tamoxifen users included in this study are presented in Table 1. The mean age of the users was 45.0 ± 9.7 years. In terms of age groups, most users were 40–49 years old (54.0%), followed by 50–59 years old (18.0%). The indications for tamoxifen use varied, with breast cancer being the most common (69.9%), followed by ductal carcinoma in situ (17.7%) and gynecomastia (10.5%). The mean duration of tamoxifen use was 36.0 ± 21.8 months, and the mean daily dose was 20.0 ± 3.0 mg.

The annual trends in the number of overall tamoxifen users and initiators between 2015 and 2021 are presented in Table 2, and the proportion of users in the entire Korean population each year is also depicted in the table. In 2015, there were 54,056 patients using tamoxifen, accounting for 0.106% of the Korean population. The number of patients who initiated tamoxifen therapy that year was 14,065, representing 0.028% of the Korean population. Over the subsequent years, both the total number of tamoxifen users and the number of patients who initiated therapy gradually increased. By 2021, the total number of patients using tamoxifen reached 81,720, comprising 0.158% of the Korean population, while 18,012 patients initiated tamoxifen therapy, accounting for 0.035% of the population. These findings demonstrate an increasing trend in the numbers of tamoxifen users and initiators over the study period, indicating an increasing population at risk of tamoxifen-induced retinal toxicity in South Korea.

### 3.2. Performance, Timing, and Modalities of Baseline and Subsequent Monitoring Examinations

Only a small proportion (6.5%) of the study population underwent baseline screening within 1 year of initiating tamoxifen therapy (Table 3). The proportion of patients who underwent ophthalmic examination at any time after tamoxifen administration was 20.4%. Appendix A shows the timing of baseline screening from the start date of tamoxifen treatment, showing a gradual decrease in percentages over time. Funduscopy/fundus photography was the most commonly used modality for baseline screening and was performed in 99.0% of cases. OCT was used in only 21.9% of patients who underwent baseline examinations. A small fraction of patients underwent automated VF (6.7%), FAF (2.8%), and FA (0.8%).

Among the patients who underwent baseline screening, 27.8% underwent subsequent monitoring examinations at any time after baseline. The mean number of monitoring examinations per year in those receiving monitoring examinations was found to be 0.68 ± 0.45, indicating a relatively low frequency of monitoring. The mean/median intervals from one examination to the subsequent examination were shortened from the baseline examination to the subsequent follow-up examinations. Similar to the baseline examinations, funduscopy/fundus photography was the most commonly employed modality for subsequent monitoring (98.6% utilization). OCT was used in 29.6% for the subsequent monitoring examinations, whereas other modalities were rarely used for baseline or monitoring examinations.

### 3.3. Trends of Retinopathy Screening among Tamoxifen Users over the Study Period

Table 4 presents the yearly trends in the proportion of tamoxifen users receiving retinopathy screening, including baseline examination and subsequent monitoring. From 2015 to 2020, the number of patients who underwent baseline examinations within 1 year gradually increased from 290 (4.6% among the annual users) in 2015 to 573 (7.7%) in 2020. Regarding subsequent monitoring, the percentage of patients examined within 6 months among those with baseline examinations ranged from 11.4% (in 2015) to 13.4% (in 2016) over the study period, while some fluctuations without a definite trend over time were noted. The percentage of patients monitored within 1 year of baseline screening among those with baseline examinations also showed a similar trend (Appendix A), with fluctuations ranging between 16.1% (in 2020) and 19.0% (in 2016).

## 4. Discussion

This study aimed to investigate the trends and patterns of retinopathy screening among tamoxifen users in South Korea. Our analyses showed the population of tamoxifen users, performance and timing of baseline and subsequent monitoring examinations, and trends in retinopathy screening over the study period. Our data indicate that the frequency of monitoring was relatively low, with funduscopy/fundus photography being the preferred modality, followed by OCT. These findings suggest that screening practices for tamoxifen retinopathy should be enhanced to ensure regular and appropriate screening.

Regarding pathogenesis, tamoxifen retinopathy shares common features with Macular Telangiectasia type 2 (MacTel) [9,10], including telangiectasia of macular blood vessels and crystalline deposits in the macula, as highlighted by various studies [11,12]. These similarities suggest potential overlapping mechanisms in the development of tamoxifen retinopathy and MacTel type 2, warranting further investigation into their shared pathophysiology. For example, both conditions may involve vascular endothelial growth factor (VEGF) pathways and Muller cell defects in their pathogenesis, as suggested by recent research [9,10,13]. In addition, emerging evidence suggests a potential link between inflammation and tamoxifen-induced retinal changes [14], although further validation is needed to confirm this association. Moreover, there is a notable paucity of studies investigating biomarkers for tamoxifen retinopathy, highlighting the importance of research efforts aimed at identifying such markers for earlier detection and improved management strategies.

Analysis of annual trends in the number of tamoxifen users and initiators revealed a gradual increase over the study period. The total number of tamoxifen users was expected to reach 81,720 by 2021, representing 0.158% of the Korean population. Similarly, the number of patients initiating tamoxifen therapy has increased, reaching 18,012 by 2021. These findings suggest a growing population at a risk of tamoxifen-induced retinal toxicity in South Korea. As the number of tamoxifen users and initiators increases, it becomes crucial to establish effective retinopathy screening programs to ensure the early detection of potential retinal toxicity.

The study population consisted of 43,848 tamoxifen users, excluding those with prior ophthalmic diseases or conditions requiring retinopathy screening (i.e., diabetes mellitus), to include those requiring toxicity screening. Most users were female, consistent with the prevalent use of tamoxifen for breast cancer treatment. The mean age of the users was 45.0 years, with the highest proportion being 40–49 years old. The mean duration of tamoxifen use in our population was 36.0 months, indicating a significant period of medication exposure that is deemed sufficient to cause tamoxifen retinopathy according to the literature [2,15,16,17]. These data underscore the clinical significance of retinopathy screening for tamoxifen users, particularly in light of the escalating prevalence of breast cancer worldwide.

Our study identified several findings regarding the performance and timing of the baseline and subsequent monitoring examinations. Only a small proportion (6.5%) of tamoxifen users received baseline screening within 1 year of initiating therapy, indicating delayed or no referral to ophthalmologists for retinopathy screening. Remarkably, the overall proportion of patients undergoing any ophthalmic examination after tamoxifen use was only 20.4%, highlighting the need for increased awareness of tamoxifen-induced retinal toxicity and screening among prescribing physicians. By disseminating knowledge about the potential ocular side effects and the need for baseline and regular screening, ophthalmologists can raise awareness among prescribing physicians and promote appropriate referrals for screening examinations.

Funduscopy/fundus photography is the most commonly used modality for baseline screening and subsequent monitoring, indicating its widespread availability and ability to detect crystalline retinopathy with sensitivity [1,18]. However, the limited utilization of OCT shown in our data (21.9% at baseline and 29.6% for subsequent monitoring) may result in the failure to detect the subtle retinal changes associated with tamoxifen use [11,15]. For instance, tamoxifen retinopathy can present with pseudocystic foveal cavitation or photoreceptor disruption in OCT [2,19,20], in addition to typical refractile crystalline depositions in fundus photographs (crystalline retinopathy). Recent studies have shown a significant role of OCT in the sensitive detection of earlier changes, such as intraretinal pseudocysts and alterations of the photoreceptor layer [2,15,19,20]. Thus, our data suggest the need for the enhanced use of OCT for tamoxifen retinopathy screening. In light of this, the establishment of expert recommendations or guidelines emphasizing the utilization of OCT for tamoxifen retinopathy screening may be useful. In contrast, other modalities such as automated VF, FAF, and FA were rarely used, possibly reflecting their minor roles in retinopathy screening compared with funduscopy or OCT for tamoxifen users [1].

The frequency of subsequent monitoring examinations was relatively low, with a mean of 0.68 monitoring examinations per year. Furthermore, the percentage of patients receiving subsequent monitoring examinations within 1 year (annual examination) after the baseline examination was less than 20%. This finding implies that the frequency of monitoring should be increased and standardized. Ophthalmologists should contribute to ensuring the timely detection of retinopathy through several measures, including regular and timely monitoring and patient education/motivation for follow-up visits. Moreover, the analysis of the yearly trend in retinopathy screening among tamoxifen users further highlights the role of ophthalmologists in improving screening practices for tamoxifen retinopathy. For instance, although the proportion of patients undergoing baseline examinations within 1 year of tamoxifen use gradually increased, the percentage of patients examined within 6 months or 1 year after baseline screening fluctuated without a clear trend over time. This indicates an improvement in the timely initiation of retinopathy screening, which is mainly determined by prescribing physicians’ referral to ophthalmologists, but no improvement in regular monitoring over time. Therefore, ophthalmologists should educate tamoxifen users about the importance of retinopathy screening, potential symptoms of retinal toxicity, and the significance of regular follow-up visits and motivate them to receive regular monitoring.

Additionally, ophthalmologists can contribute to the development and dissemination of evidence-based guidelines for screening for tamoxifen retinopathy. Specific guidelines may ensure consistent and standardized screening practices by ophthalmologists. For hydroxychloroquine, another well-known drug causing toxic retinopathy, recommendations for screening established by multiple organizations play significant roles in standardized screening practices and sensitive detection of the retinal toxicity [21,22]. In our previous study on hydroxychloroquine, there were significant improvements in timely baseline and annual monitoring over the same study period [8]. We believe that the difference in the presence of established guidelines between tamoxifen and hydroxychloroquine retinopathy might have led to the difference in the trend of screening practices over time between the two.

More specifically, for consistency and frequency of tamoxifen retinopathy screening, it is essential to establish a standardized protocol. This protocol should entail regular ophthalmologic evaluations for patients undergoing tamoxifen therapy, with initial screening at treatment initiation and subsequent follow-up examinations scheduled at regular intervals thereafter. Although the timing and frequency of screenings may vary based on factors such as treatment duration (or cumulative dosage) and individual risk factors, there is currently no consensus on the optimal schedule. However, a common recommendation is to conduct regular screenings including OCT, perhaps every 6 months, particularly for patients who have been on tamoxifen at a dosage of 20 mg/day for at least 2 years [1]. More frequent assessments may be warranted for individuals with higher risk profiles, preexisting ocular conditions, or symptomatic presentations. Effective collaboration between oncologists and ophthalmologists is imperative to ensure comprehensive and timely monitoring in patients undergoing tamoxifen therapy.

At present, there is no established consensus on the optimal treatment for tamoxifen-induced retinal toxicity. Discontinuing tamoxifen therapy should be considered as a preventative measure against further retinal damage. However, one promising approach involves the use of intravitreal steroids or anti-VEGF agents, similar to treatments employed for MacTel type 2 [1,23]. These medications have shown potential in alleviating pathological changes such as the cystic changes and macular edema commonly observed in tamoxifen retinopathy [1,24]. Additionally, it is important to address comorbidities such as hyperlipidemia and elevated body mass index as these have been linked to a higher risk of retinal changes with tamoxifen use [2]. Therefore, lowering lipid levels, promoting a low-fat diet and lifestyle modifications, might be beneficial in reducing the risk of retinal toxicity. Overall, further research is needed to fully understand the efficacy and safety of potential treatment modalities for tamoxifen-induced retinal toxicity and to develop novel therapies.

Although our study provides valuable insights into the trends and patterns of retinopathy screening among tamoxifen users, several limitations should be acknowledged when interpreting the results. First, it was conducted in South Korea, which may limit the generalizability of the findings to other populations with potentially different healthcare systems, cultural practices, and tamoxifen utilization patterns. Therefore, caution should be exercised when extrapolating these results to other populations. Second, our study relied on retrospective data obtained from the health claims database, which may include inherent limitations, such as missing or incorrect information, particularly in the diagnosis codes. Additionally, the study lacked information on the specific indications for ophthalmic examinations, which could have impacted the frequency and modality choices [25]. Finally, the study did not explore the reasons behind the observed trends in retinopathy screening or the potential barriers to screening practices. Understanding the factors influencing screening practices and identifying potential barriers to timely and adequate monitoring are crucial for developing targeted interventions and improving screening rates among tamoxifen users.

Another significant limitation of our study is the absence of an analysis of tamoxifen retinopathy cases. The lack of a specific diagnostic code for toxic maculopathy in the eighth revision of the Korean Standard Classification of Diseases hampered our ability to accurately identify all instances of tamoxifen retinopathy, potentially introducing bias. Moreover, tamoxifen retinopathy can manifest with diverse features like macular edema, which further complicates the interpretation of diagnostic codes. Although including the code for macular edema might enhance the detection of tamoxifen retinopathy cases, it also raises the risk of bias due to the other diverse causes of macular edema. Therefore, we intended to focus our analyses on investigating practice patterns for retinal toxicity and drug usage among tamoxifen users in this study rather than determining the precise incidence or prevalence of tamoxifen retinopathy.

## 5. Conclusions

Our study provides insights into the population of tamoxifen users, the performance of baseline and subsequent monitoring examinations, and trends in retinopathy screening over time. These findings highlight the need for continued efforts to optimize screening protocols, increase awareness among prescribing physicians and tamoxifen users, and improve the consistency and frequency of retinopathy screening in this at-risk population.

## Figures and Tables

**Figure 1 jcm-13-02167-f001:**
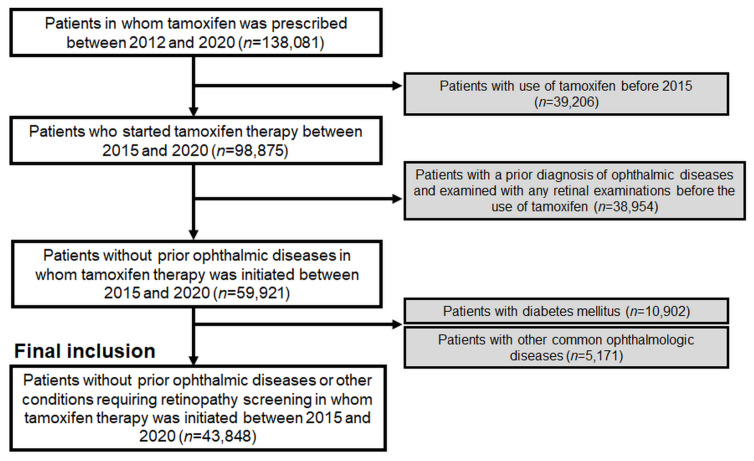
A flowchart of the study population and inclusion/exclusion criteria in this study.

**Table 1 jcm-13-02167-t001:** Demographic and clinical information of the tamoxifen users included in this study.

Characteristics	Overall Users (*n* = 43,848)
**Sex**	
Male	5003 (11.4%)
Female	38,845 (88.6%)
**Mean age (±SD), years**	45.0 ± 9.7
<20	688 (1.6%)
20–29	1860 (4.2%)
30–39	6968 (15.9%)
40–49	23,667 (54.0%)
50–59	7903 (18.0%)
60–69	1945 (4.4%)
≥70	817 (1.9%)
**Indication for tamoxifen use**	
Breast cancer	30,629 (69.9%)
Ductal carcinoma in situ	7744 (17.7%)
Gynecomastia	4606 (10.5%)
Others	869 (2.0%)
**Mean duration of tamoxifen use (±SD), months**	36.0 ± 21.8
Less than 1 year	7858 (17.9%)
1–2 years	6287 (14.3%)
2–3 years	7857 (17.9%)
3–4 years	6849 (15.6%)
4–5 years	8715 (19.9%)
5 years or longer	6282 (14.3%)
**Mean daily dose of tamoxifen (±SD), mg/day**	20.0 ± 3.0
Less than 15 mg	1064 (2.4%)
15–20 mg	808 (1.8%)
20–25 mg	41,261 (94.1%)
25 mg or greater	715 (1.6%)

SD, standard deviation.

**Table 2 jcm-13-02167-t002:** Annual number of tamoxifen users and those who started tamoxifen therapy between 2015 and 2021.

Year	Total Number of Patients Using Tamoxifen (% among Entire Korean Population in Each Year ^†^)	Annual Number of Patients Who Initiated Tamoxifen Therapy (% among Korean Population ^†^)
2015	54,056 (0.106%)	14,065 (0.028%)
2016	59,426 (0.116%)	15,690 (0.031%)
2017	64,760 (0.126%)	16,689 (0.032%)
2018	69,957 (0.136%)	17,272 (0.033%)
2019	74,860 (0.145%)	17,990 (0.035%)
2020	77,943 (0.150%)	17,169 (0.033%)
2021	81,720 (0.158%)	18,012 (0.035%)

^†^ Obtained by dividing the number of tamoxifen users by that of the entire Korean population in each year (from 51,014,947 in 2015 to 51,744,876 in 2021).

**Table 3 jcm-13-02167-t003:** Descriptive statistics of the timing and modalities used for the baseline examination (1st ophthalmic examination after tamoxifen use) and monitoring (subsequent follow-up screening) among all patients between 2015 and 2021.

Characteristics	Value
**Timing**	
No. of patients receiving any ophthalmic examination after tamoxifen use/No. of users (%)	8961/43,848 (20.4%)
No. of patients receiving ophthalmic examination within 1 year of tamoxifen use/No. of users (%)	2836/43,848 (6.5%)
No. of patients receiving any subsequent monitoring examination/No. of patients receiving baseline screening (%)	2492/8961 (27.8%)
No. of monitoring examinations per year after baseline ones, numbers/year	0.68 ± 0.45
Timing of the baseline examination since tamoxifen use, median (Q1–Q3), days	645 (280–1161)
Mean/median (Q1–Q3) interval between baseline examination and 1st monitoring exam, months	12.5 ± 14.2/7.1 (1.2–19.3)
Mean/median (Q1–Q3) interval of monitoring between 1st and 2nd monitoring exam, months	6.9 ± 8.7/3.4 (0.7–10.1)
Mean/median (Q1–Q3) interval of monitoring between 2nd and 3rd monitoring exam, months	5.8 ± 8.4/2.8 (0.6–7.1)
**Modalities used** Funduscopy/fundus photography Optical coherence tomography Automated visual fields Fundus autofluorescence Fluorescein angiography Others	Baseline/Monitoring (%) 8873 (99.0%)/2458 (98.6%) 1960 (21.9%)/738 (29.6%) 602 (6.7%)/205 (8.2%) 253 (2.8%)/138 (5.5%) 70 (0.8%)/52 (2.1%) 270 (3.0%)/162 (6.5%)

**Table 4 jcm-13-02167-t004:** Yearly trends in the proportion of tamoxifen users undergoing retinopathy screening (baseline examination and subsequent monitoring).

Year	Baseline Examination within 1 Year	Monitoring	
		Examined within 6 Months from Baseline Exam (% among Those with Baseline)	Examined within 1 Year from Baseline Exam (% among Those with Baseline)
2015	290 (4.6%)	210 (11.4%)	286 (16.3%)
2016	411 (5.7%)	252 (13.4%)	358 (19.0%)
2017	453 (6.0%)	225 (12.9%)	302 (17.4%)
2018	524 (6.9%)	196 (13.2%)	268 (18.0%)
2019	585 (7.4%)	150 (11.7%)	221 (17.3%)
2020	573 (7.7%)	102 (12.5%)	131 (16.1%)

## Data Availability

Data are unavailable due to privacy and ethical restrictions.

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
