# Peer review of "Nationwide Screening Practices for Tamoxifen Retinal Toxicity in South Korea: A Population-Based Cohort Study"

_jcm, 2024, doi:10.3390/jcm13082167_

Round 1

Reviewer 1 Report

Comments and Suggestions for Authors

Nationwide Screening Practices of Tamoxifen Retinopathy in South Korea: a Population-Based Cohort Study

Tamoxifen, a selective estrogen receptor (ER) modulator, is widely used to treat and prevent hormone receptor-positive breast cancer. One of the adverse reactions of tamoxifen is that it can cause bilateral pigmentary retinopathy severe enough to discontinue. the treatment. Tamoxifen and toremifene can also cause macular crystals, macular drusen, and yellowish spots in the macula area.

The topic presented by the authors is very interesting and very worrying for the pharmacological use of tamoxifen as anti-inflammatory agents and its adverse reactions.

After reading the complete manuscript, I have observed that the literature consulted for this manuscript is very short, only presenting 17 citations.

In relation to these observations, I have a few questions:

1) What are the inflammatory biomarkers of tamoxifen retinal toxicity observed during the study?

2) What protocols do you recommend for the consistency and frequency of retinopathy screening of tamoxifen use in South Korea?

3) Can you explain the promising strategy for the treatment of retinal toxicity of tamoxifen?

Author Response

Reviewer 1 

Tamoxifen, a selective estrogen receptor (ER) modulator, is widely used to treat and prevent hormone receptor-positive breast cancer. One of the adverse reactions of tamoxifen is that it can cause bilateral pigmentary retinopathy severe enough to discontinue. the treatment. Tamoxifen and toremifene can also cause macular crystals, macular drusen, and yellowish spots in the macula area.

The topic presented by the authors is very interesting and very worrying for the pharmacological use of tamoxifen as anti-inflammatory agents and its adverse reactions.

After reading the complete manuscript, I have observed that the literature consulted for this manuscript is very short, only presenting 17 citations.

→ Thank you very much for your thoughtful review and helpful suggestions. In response to this comment, we have performed more extensive literature search and found several additional relevant references for Introduction and Discussion sections. We have added 8 more citations (now 25 in total) into the References section.

In relation to these observations, I have a few questions:

1) What are the inflammatory biomarkers of tamoxifen retinal toxicity observed during the study?

→ This is a very interesting point, as inflammation and tamoxifen retinal toxicity may indeed be associated, although this link remains unproven. Previous research has uncovered intriguing findings in two separate mouse models of photoreceptor degeneration, where tamoxifen surprisingly exhibited potent neuroprotective effects. These effects were correlated with reduced microglial activation and inflammatory cytokine production in the retina in vivo, as well as a decrease in microglia-mediated toxicity to photoreceptors in vitro. Thus, it is plausible that tamoxifen retinal toxicity may not be solely mediated by inflammation, although this hypothesis requires further investigation. Notably, no specific inflammatory biomarkers have been identified for tamoxifen retinal toxicity. In our Discussion section, we have addressed the potential association between inflammation and the pathogenesis of tamoxifen retinal toxicity, suggesting avenues for future research to identify inflammatory biomarkers in tamoxifen retinopathy.

In the text, we have added the sentences “In addition, emerging evidence suggests a potential link between inflammation and tamoxifen-induced retinal changes, although further validation is needed to confirm this association. Moreover, there is a notable paucity of studies investigating biomarkers for tamoxifen retinopathy, highlighting the importance of research efforts aimed at identifying such markers for earlier detection and improved management strategies.” (Page 7, Lines 214-219)

2) What protocols do you recommend for the consistency and frequency of retinopathy screening of tamoxifen use in South Korea?

→ Thank you very much for your comment. For consistency and frequency of retinopathy screening in South Korea, implementing a standardized protocol is required for Korean patients taking tamoxifen. This protocol should include regular ophthalmologic evaluations for patients on tamoxifen therapy, with initial screening at the commencement of treatment and subsequent follow-up examinations at regular intervals thereafter. The timing and frequency of screening may vary depending on factors such as the duration of tamoxifen therapy, cumulative dose, and individual risk factors but there has been no consensus over the timing and frequencies. However, a common recommendation is to conduct regular screenings including OCT, perhaps every 6 months, in patients who have taken tamoxifen 20 mg/day for at least 2 years, but with more frequent assessments for those at higher risk or preexisting ocular conditions or for those with symptoms. Collaboration between oncologists and ophthalmologists is also essential to ensure comprehensive and timely monitoring of retinal health in patients receiving tamoxifen therapy. We have added these points in the Discussion section as follows:

More specifically, for consistency and frequency of tamoxifen retinopathy screening, it is essential to establish a standardized protocol. This protocol should entail regular ophthalmologic evaluations for patients undergoing tamoxifen therapy, with initial screening at treatment initiation and subsequent follow-up examinations scheduled at regular intervals thereafter. Although the timing and frequency of screenings may vary based on factors such as treatment duration, cumulative dosage, and individual risk factors, there is currently no consensus on the optimal schedule. However, a common recommendation is to conduct regular screenings including OCT, perhaps every 6 months, particularly for patients who have been on tamoxifen 20 mg/day for at least 2 years. More frequent assessments may be warranted for individuals with higher risk profiles, preexisting ocular conditions, or symptomatic presentations. Effective collaboration between oncologists and ophthalmologists is imperative to ensure comprehensive and timely monitoring in patients undergoing tamoxifen therapy. (Page 9, Lines 294-306)

3) Can you explain the promising strategy for the treatment of retinal toxicity of tamoxifen?

→ Thank you for bringing up this important point. Unfortunately, there is currently no consensus on the optimal treatment approach for tamoxifen-induced retinal toxicity. However, one promising strategy involves the use of intravitreal steroids or anti-VEGF agents, similar to treatments used for MacTel Type 2. These medications have shown potential in alleviating pathological changes such as cystic changes and macular edema commonly observed in tamoxifen retinopathy. Additionally, it is important to address comorbidities such as hyperlipidemia and elevated body mass index, as these have been linked to a higher risk of retinal changes with tamoxifen use. Therefore, lowering lipid levels, promoting a low-fat diet and lifestyle modifications, might be beneficial in reducing the risk of retinal toxicity. Overall, further research is needed to fully understand the efficacy and safety of potential treatment modalities for tamoxifen-induced retinal toxicity and to develop novel therapies. These points have been incorporated into the Discussion section as follows:
At present, there has been no established consensus on the optimal treatment for tamoxifen-induced retinal toxicity. Discontinuing tamoxifen therapy should be con-sidered as a preventative measure against further retinal damage. However, one promising approach involves the use of intravitreal steroids or anti-VEGF agents, sim-ilar to treatments employed for MacTel Type 2.[1,23] These medications have shown potential in alleviating pathological changes such as cystic changes and macular ede-ma commonly observed in tamoxifen retinopathy.[1,24] Additionally, it is important to address comorbidities such as hyperlipidemia and elevated body mass index, as these have been linked to a higher risk of retinal changes with tamoxifen use.[2] Therefore, lowering lipid levels, promoting a low-fat diet and lifestyle modifications, might be beneficial in reducing the risk of retinal toxicity. Overall, further research is needed to fully understand the efficacy and safety of potential treatment modalities for tamoxi-fen-induced retinal toxicity and to develop novel therapies. (Page 9, Lines 307-319)

Reviewer 2 Report

Comments and Suggestions for Authors

This is a large population study addressing an underexplored topic. 

Methods need major revisions.

The definitions are not properly presented. The criteria for diagnosing tamoxifen retinopathy and the distribution of maculopathy/ macular oedema among patients need clearer definition, including specifics on unilateral versus bilateral cases. Adjustments in statistical analyses to accommodate bilateral cases using mixed models are necesary to take into account the violation of independence assumption. 

Author Response

Reviewer 2 

This is a large population study addressing an underexplored topic. 

Methods need major revisions.

The definitions are not properly presented. The criteria for diagnosing tamoxifen retinopathy and the distribution of maculopathy/ macular oedema among patients need clearer definition, including specifics on unilateral versus bilateral cases.

Adjustments in statistical analyses to accommodate bilateral cases using mixed models are necesary to take into account the violation of independence assumption. 

→ We deeply appreciate your review and valuable feedback. In our study, we aimed to evaluate nationwide tamoxifen use based on prescription records and practice patterns among patients receiving tamoxifen therapy. As such, our focus was on the population-level analysis of practice patterns for patients taking tamoxifen rather than on evaluating individual eyes with tamoxifen retinopathy, which would require distinguishing between unilateral and bilateral cases of the condition. Therefore, the diagnostic criteria for tamoxifen retinopathy and the data on the distribution of maculopathy or macular edema were not used for this study. Further, adjustments for bilateral cases using methods such as linear mixed models were not necessary in our study design. We appreciate your suggestion and believe that these aspects should be carefully addressed in future studies focusing specifically on patients with tamoxifen retinopathy.

Reviewer 3 Report

Comments and Suggestions for Authors

The letters in Fig.1 are a little bit vague. Please refine and improve it.

Author Response

Reviewer 3 

The letters in Fig.1 are a little bit vague. Please refine and improve it.

→ Thank you very much for your review and your valuable feedback on Figure 1. We have refined by unnecessary parts and improve the clarity of the letters to enhance the overall presentation of the figure.

Reviewer 4 Report

Comments and Suggestions for Authors

I consider this study to have valuable data that would be of interest if published. The major strentgh of this study is the high number of observations and large size of the study group. The major limitation is the lack of the analysis of tamoxifen retinopathy cases. Please add an analysis of the number of cases of tamoxifen retinopathy in relation to the number of people examined and the entire study group. Then, please update the results and discussion sections with the findings from this new analysis.

Author Response

Reviewer 4 

I consider this study to have valuable data that would be of interest if published. The major strentgh of this study is the high number of observations and large size of the study group. The major limitation is the lack of the analysis of tamoxifen retinopathy cases. Please add an analysis of the number of cases of tamoxifen retinopathy in relation to the number of people examined and the entire study group. Then, please update the results and discussion sections with the findings from this new analysis.

→ Thank you for highlighting this important point. We acknowledge that there are indeed several challenges in analyzing cases of tamoxifen retinopathy based on diagnostic codes from our database. Firstly, the absence of a specific diagnostic code for toxic maculopathy in the 8th revision of the Korean Standard Classification of Diseases (KCD) limits our ability to accurately capture all cases of tamoxifen retinopathy, leading to potential bias. Additionally, tamoxifen retinopathy can present with diverse features such as macular edema, complicating the interpretation of diagnostic codes. While adding the code for macular edema may increase the capture of tamoxifen retinopathy cases, it also introduces the risk of bias due to the presence of other causes of macular edema. By carefully considering these challenges, we directed our attention towards examining practice patterns for retinal toxicity and drug usage among tamoxifen users in this study, rather than calculating the exact incidence or prevalence of tamoxifen retinopathy. In response to your comment, we have addressed it in the Discussion section. Thank you for highlighting this important aspect and limitation of our study.

Another significant limitation of our study is the absence of an analysis of tamoxifen retinopathy cases. The lack of a specific diagnostic code for toxic maculopathy in the 8th revision of the Korean Standard Classification of Diseases hampers our ability to accurately identify all instances of tamoxifen retinopathy, potentially introducing bias. Moreover, tamoxifen retinopathy can manifest with diverse features like macular edema, which further complicates the interpretation of diagnostic codes. Although in-cluding the code for macular edema might enhance the detection of tamoxifen reti-nopathy cases, it also raises the risk of bias due to other diverse causes of macular edema. To address these challenges and potential biases, we intended to focus our analyses on investigating practice patterns for retinal toxicity and drug usage among tamoxifen users in this study, rather than determining the precise incidence or preva-lence of tamoxifen retinopathy. (Page 9, Line 335 – Page 10, Line 346)

Round 2

Reviewer 2 Report

Comments and Suggestions for Authors

The authors partially addressed my concerns. 

1) They need to clarify in the revised manuscript the fact that they did not distinguish patients into bilateral and unilateral cases. 

2) The definitions are still not properly presented. The criteria for diagnosing tamoxifen retinopathy and the distribution of maculopathy/ macular oedema among patients need clearer definition. 

Author Response

The authors partially addressed my concerns. 

1) They need to clarify in the revised manuscript the fact that they did not distinguish patients into bilateral and unilateral cases. 

-> Thank you for your review. Our study aims to evaluate nationwide tamoxifen use (specifically the number of tamoxifen users) based on prescription records and screening practice patterns. We are not assessing tamoxifen retinopathy incidence or prevalence, in which defining tamoxifen retinopathy is crucial. Similar methodologies were employed in our previous papers on hydroxychloroquine (Kim et al. JAMA Network Open, 2023) and pentosan polysulfate (PPS) users (Kim et al. Ophthalmology Retina, 2024) where retinopathy definitions were not provided as they didn't align with study objectives. As our focus is on tamoxifen users, not on patients with tamoxifen retinopathy, discriminating between bilateral or unilateral cases is unnecessary. For studies on tamoxifen retinopathy prevalence or incidence, clear definitions on the retinopathy and bilateral/unilateral cases are essential. While we appreciate your suggestion, our current study does not delve into such clinical information on tamoxifen retinopathy.

  2) The definitions are still not properly presented. The criteria for diagnosing tamoxifen retinopathy and the distribution of maculopathy/ macular oedema among patients need clearer definition.    -> As mentioned above, for studies assessing the clinical parameters in patients with tamoxifen retinopathy or incidence/prevalence of tamoxifen retinopathy, clear definitions for tamoxifen retinopathy are essential. While we value your suggestion greatly, our current research does focus not on tamoxifen retinopathy but on tamoxifen users and ophthalmologic examinations performed for screening in the users. However, we do plan to undertake a separate study specifically on tamoxifen retinopathy incidence or prevalence, with a differentiation between macular edema and other forms. Your input on the definitions will be thoroughly taken into account for that study. Thank you once again for your valuable suggestion.